# Advances in Antitumor Effects Using Liposomal Citrinin in Induced Breast Cancer Model

**DOI:** 10.3390/pharmaceutics16020174

**Published:** 2024-01-26

**Authors:** Michely Laiany Vieira Moura, Ag-Anne Pereira Melo de Menezes, José Williams Gomes de Oliveira Filho, Maria Luiza Lima Barreto do Nascimento, Antonielly Campinho dos Reis, Alessandra Braga Ribeiro, Felipe Cavalcanti Carneiro da Silva, Adriana Maria Viana Nunes, Hercília Maria Lins Rolim, Ana Amélia de Carvalho Melo Cavalcante, João Marcelo de Castro e Sousa

**Affiliations:** 1Laboratory of Toxicological Genetics—LAPGENIC, Graduate Program in Pharmaceutical Sciences, Federal University of Piauí, Teresina 64049-550, Brazil; michelylaiany@gmail.com (M.L.V.M.); ag-anne@hotmail.com (A.-A.P.M.d.M.); williamsfilho@ifpi.edu.br (J.W.G.d.O.F.); mlbrreto95@gmail.com (M.L.L.B.d.N.); antonielyreis@gmail.com (A.C.d.R.); felipecavalcanticarneiro@gmail.com (F.C.C.d.S.); anameliamelocavalcante@gmail.com (A.A.d.C.M.C.); j.marcelo@ufpi.edu.br (J.M.d.C.e.S.); 2CBQF—Centro de Biotecnologia e Química Fina—Laboratório Associado, Escola Superior de Biotecnologia, Universidade Católica Portuguesa, Rua Diogo Botelho 1327, 4169-005 Porto, Portugal; alessandra.bragaribeiro@gmail.com; 3Department of Biophysics and Physiology, Federal University of Piauí, Teresina 64049-550, Brazil; adriananunes@ufpi.edu.br; 4Laboratory of Pharmaceutical Nanosystems—NANOSFAR, Graduate Program in Pharmaceutical Sciences, Federal University of Piauí, Teresina 64049-550, Brazil

**Keywords:** fungal metabolites, cytotoxicity, nanotechnology, breast cancer

## Abstract

The study aimed to evaluate the antitumor and toxicogenetic effects of liposomal nanoformulations containing citrinin in animal breast carcinoma induced by 7,12-dimethylbenzanthracene (DMBA). *Mus musculus* virgin females were divided into six groups treated with (1) olive oil (10 mL/kg); (2) 7,12-DMBA (6 mg/kg); (3) citrinin, CIT (2 mg/kg), (4) cyclophosphamide, CPA (25 mg/kg), (5) liposomal citrinin, LP-CIT (2 μg/kg), and (6) LP-CIT (6 µg/kg). Metabolic, behavioral, hematological, biochemical, histopathological, and toxicogenetic tests were performed. DMBA and cyclophosphamide induced behavioral changes, not observed for free and liposomal citrinin. No hematological or biochemical changes were observed for LP-CIT. However, free citrinin reduced monocytes and caused hepatotoxicity. During treatment, significant differences were observed regarding the weight of the right and left breasts treated with DMBA compared to negative controls. Treatment with CPA, CIT, and LP-CIT reduced the weight of both breasts, with better results for liposomal citrinin. Furthermore, CPA, CIT, and LP-CIT presented genotoxic effects for tumor, blood, bone marrow, and liver cells, although less DNA damage was observed for LP-CIT compared to CIT and CPA. Healthy cell damage induced by LP-CIT was repaired during treatment, unlike CPA, which caused clastogenic effects. Thus, LP-CIT showed advantages for its use as a model of nanosystems for antitumor studies.

## 1. Introduction

Breast cancer (BC) is a complex disease that presents a high degree of heterogeneity [1,2]. BC presents variable morphological and biological characteristics and, therefore, differences in clinical behavior and response to treatment [3]. It is estimated that 1.6 million BC cases occur worldwide each year, making it the most common cancer in women. Currently, surgical resection, radiotherapy, hormone therapy, and mainly chemotherapy represent the main treatment options in the early BC stages [4].

In BC treatment, the antineoplastic agents doxorubicin, cyclophosphamide, paclitaxel, carboplatin, and cisplatin are traditionally used [5]. However, the development of drug resistance and significant side effects have weakened the effectiveness of these therapies [6]. Antineoplastic drugs are limited by their limited therapeutic window, non-selectivity, and high toxicity risk. Therefore, minimizing chemotherapy-induced side effects through selective tumor delivery through nanoformulations would increase their efficiency [7].

Nanoparticles present excellent properties, such as a small size, large surface area, high surface reactivity, unique physicochemical properties, high surface-to-volume ratio, and superior reactivity over their volume counterparts. These features give them unique physical properties, resulting in novel opportunities for early detection, efficiency, and cancer diagnosis [8,9]. Tumor-targeted drug delivery systems such as liposomes have emerged as an essential advance in cancer treatment, ensuring sufficient drug levels accumulating at the tumor site, in addition to the possibility of enhancing the cytotoxicity and antitumor effects of natural products such as citrinin [10].

Citrinin (CIT) is a secondary metabolite of fungi of the genera *Penicillium* (*P. citrinum*, *P. expansum*, *P. verrucosum*, and *P. camamberti*) and *Aspergillus* (*A. terreus* and *A. niveus*) [11]. The mechanism of CIT toxicity implies disturbances in mitochondrial permeability transition, calcium flux, the electron release system, and cytochrome C; it induces mutagenic changes, as well as oxygen reactive species (ROS) formation [12,13,14,15]. These CIT biological effects, e.g., cytotoxic and genotoxic, are related to the effects of antineoplastic drugs, as they are apoptosis induction mechanisms [16]. Therefore, citrinin presents potential to be a good candidate for antineoplastic agents, since it induces cytotoxic, genotoxic, and mutagenic effects in cancer cells, blocking carcinogenesis.

In this context, considering the constant search for novel antitumor drugs and better delivery systems, the present study aimed to incorporate the fungal metabolite citrinin into a liposomal nanosystem, evaluating its toxic, genotoxic, and antitumor effects in an animal model of breast cancer induced by 7,12-dimethylbenzanthracene (DMBA).

## 2. Materials and Methods

### 2.1. Chemical Reagents

The chemical carcinogen 7,12-dimethylbenzanthracene (DMBA), the antineoplastic cyclophosphamide (CPA), citrinin (CIT) with purity of 98%, and all other chemicals and reagents necessary to carry out this study were purchased from Sigma-Aldrich (Chem Ex. Co., St. Louis, MO, USA).

### 2.2. Liposomal Citrinin (LP-CIT)

Positively charged liposomal citrinin (LP-CIT) was prepared by the lipid film hydration method according to Anselem [17], with adaptations. LP-CIT presented pH 7.4, a particle diameter of 143.3 nm, a polydispersion index (PDI) of 0.3, and zeta potential of +28.5 mV. The encapsulation efficiency of the method used was 63.82% citrinin. The stability of this system in an aqueous medium at 4 °C was 21 days.

### 2.3. Animals and Treated Groups

The animals were obtained from the Central Animal Facility of the Center for Agricultural Sciences (CCA), at the Federal University of Piauí. For the toxicogenetic monitoring of DMBA, female mice (*Mus musculus*), virgin, Swiss, albino, 6 to 7 weeks old, and with a weight between 25 and 30 g, were subjected to acclimatization for one week, under monitored temperature conditions equivalent to 25 ± 1 °C and a 12-h light/dark cycle, with free access to pellets (Purina^®^) and water. The animals were divided into six groups: (1) olive oil (olive oil 10 mL/kg, v.o.); (2) DMBA (6 mg/kg, v.o.); (3) free citrinin—CIT (2 μg/kg, p.o.); (4) cyclophosphamide—CPA (25 mg/kg, i.p.) [18]; (5) LP-CIT (2 µg/kg, i.p.); (6) LP-CIT (6 µg/kg, i.p.). All experiments were previously approved by the Ethics Committee on Animal Experimentation of the Federal University of Piauí (No. 469/18, ANNEX B).

### 2.4. Breast Cancer Induced by 7,12-Dimethylbenzanthracene (DMBA)

The mammary carcinogenesis assay was conducted according to the protocol described by Alencar [19], with adaptations. DMBA was dissolved in olive oil and administered by gavage at a dose of 6 mg/kg (weekly). The animals were divided into seven groups (*n* = 5/group). Group 1 was treated only with the olive oil vehicle (10 mL/kg, v.o.); the other groups were treated with DMBA (6 mg/kg, p.o.) for 11 weeks. All animals were weighed, and the individual dose was calculated based on weight [20]. Tumor formation was monitored, for 11 weeks, by physical palpation and diameter measurement with the aid of a digital caliper. After BC detection in the groups (*n* = 5/group), DMBA administration was interrupted and the animals started to receive therapy with free CIT (2 μg/kg, p.o.), CPA (25 mg/kg, i.p.), LP-CIT (2 µg/kg, i.p.), or LP-CIT (6 µg/kg, i.p.) once a week for a period of 3 weeks. All drug concentrations were established according to previous studies carried out [18]. After treatment, the mammary glands were dissected and subjected to complete necropsy, for macro- and microscopic evaluation.

The products of mammary resections and necropsy were sent for macroscopic examination (determination of dimensions and alterations in the cut surface), fixation in 10% buffered formalin, staining with hematoxylin and eosin, and automatic histotechnical processing, for paraffin inclusion. Moreover, mammary tissues and peripheral blood were collected and subjected to genotoxicity studies.

### 2.5. Toxicological Monitoring during Breast Cancer Induction and Therapy

#### 2.5.1. Hippocratic Analysis

Animals were monitored for possible changes in general (Hippocratic) activity, including vocal frenulum, irritability, touch response, writhing, righting reflex, muscle tone, gripping strength, ataxia, ear reflex, corneal reflex, tremors, convulsions, Straub syrup, hypnosis, anesthesia, lacrimation, palpebral ptosis, urination, defecation, piloerection, hypothermia, breathing, cyanosis, hyperemia, and death, after 30 min and 1 h of weekly administration of DMBA and therapy with CIT, CPA, and LP-CIT.

#### 2.5.2. Open Field

The open field test comprised an acrylic box (transparent walls and black floor, with a diameter of 40 × 15 cm^2^ in height), divided into nine squares of equal dimensions [21]. After 30 min and 1 h of each administration, each animal was placed in the apparatus, individually, and the number of crossings with the four paws by the divisions of the apparatus (spontaneous locomotor activity), the frequency of self-cleaning behavior, and the number of paw lifts were evaluated, without leaning on the apparatus walls. After each individual evaluation, the equipment was aseptically cleaned with 70% alcohol. Each week of treatment, the field was repeated.

#### 2.5.3. Rota Rod

After an interval of 5 min, the animals tested in the open field were subjected to the rota rod. The mice were placed with all four paws on a rotating stainless steel cylindrical bar, 25 mm in diameter, at a rotation speed of 17 rpm, for a period of 3 min. The time spent on the bar in seconds (TS) and the number of falls (NF) were measured in triplicate [22]. At each week of treatment, the rota rod test was repeated.

#### 2.5.4. Biochemical and Hematological Analysis

For hematological evaluation, the number of erythrocytes, hemoglobin content, hematocrit, mean corpuscular volume (MCV), mean corpuscular hemoglobin (MCH), and mean corpuscular hemoglobin concentration (MCHC) were determined as red series, while the numbers of leukocytes, neutrophils, lymphocytes, eosinophils, and monocytes were determined in white series, in addition to platelets. Blood collected with anticoagulant, directly from the heart, was analyzed using an automatic hematological device (Advia 120/Hematology Siemens). For biochemical analysis, blood was collected from the heart, immediately after euthanization, and centrifuged at 4000× *g*, for 5 min (at 4 °C). The plasma was subjected to an automatic biochemical analysis, in a Labmax 240 device, with commercial Labtest^®^ kits for urea, creatinine, aspartate aminotransferase (AST), and alanine aminotransferase (ALT).

#### 2.5.5. Euthanasia and Organ Collection

After 11 weeks of DMBA treatment and 3 weeks (12th, 13th, and 14th) of drug therapy in the groups with CIT, CP, and LP-CIT, the animals were sacrificed by overdose of anesthetic solution (sodium pentobarbital + xylazine [1:1], i.p.). Immediately, the right and left breasts, liver, and kidney were separated in an ice box. The organs were washed with a phosphate–saline buffer solution (PBS, pH 7.4), weighed, and preserved in 10% formalin solution for 24 h and later maintained in a 70% alcohol solution.

#### 2.5.6. Histopathological Evaluation

The breast tissue fragments were fixed in 3.5% formaldehyde in phosphate buffer (pH 7.6), simultaneously with 1% sucrose solution, and stored for 12 h at 4 °C, until use. Fixation was processed for 20 h, followed by dehydration of the pieces in increasing dilutions of ethanol/water (50, 70, 90, and 95%, *v*/*v*). The dry material was immersed in xylene PA and washed twice in paraffin and, subsequently, paraffin-embedded [23]. The blocks were sectioned (4 μM), in a microtome, and the sections distended in slides. Consequently, the distended sections were deparaffinized in three successive xylene PA baths and dehydrated in decreasing ethanol dilutions, as mentioned above. Soon after, the slides were stained with hematoxylin and eosin [24], and, finally, after washing and drying, the slides were covered with coverslips with the aid of Canada balsam for further analysis.

### 2.6. Genotoxic Damage, DNA Repair, and Mutagenic Profile in Tumor and Healthy Cells

#### 2.6.1. Comet Alkaline Assay

The alkaline version of the comet assay was performed, as described by Speit and Hartmann [25]. Peripheral blood (1st, 12th, 13th, 14th, and 15th weeks), liver cells, bone marrow, and mammary tissue were collected for genotoxic profile analysis. Aliquots of 10 μL of peripheral blood and 10 μL of cell suspension from different tissues (0.5 × 10^6^ cells/mL) were collected and mixed with a thin layer of low-melting-point 0.75% agarose (90 μL) and placed on slides pre-coated with 1.5% ultrapure agarose. The slides were immersed in a lysis solution (2.5 M NaCl, 100 mM EDTA, and 10 mM Tris, pH 10, with the addition of 1% Triton X-100 and 10% DMSO at the time of use), for up to 72 h at 4 °C. Then, the slides were incubated in an alkaline buffer (300 mM NaOH and 1 mM EDTA, pH > 13) for 20 min, and then immediately exposed to an electric current of 300 mA and 25 V (0.90 V/cm), for 20 min in an electrophoresis device. After electrophoresis, the slides were neutralized with 0.4 M Tris buffer, pH 7.5, and stained with silver solution. The results were expressed as the damage index (DI) and frequency of damage (FD). DI was calculated using the formula DI = Σ (number of cells in a given damage class × damage class), which ranged from 0 to 400, and FD by the following formula: FD = 100—number of class 0 cells.

#### 2.6.2. Micronucleus Test and Erythrocyte Evaluation

The mutagenic evaluation was performed according to Moraes [26]. Briefly, cells obtained from breast and liver maceration, in addition to bone marrow cells, were added with 0.3 mL of fetal bovine serum and spread on different slides. Furthermore, the peripheral blood sample was spread on smear slides. The swabs were dried for 30 min at room temperature and fixed in methanol for 10 min. Slides were then stained with Giemsa (Merck) in 0.2 mol/L phosphate buffer (pH 5.8). The micronuclei were counted in 2000 cells by photomicrography at 1000× magnification. The erythrocyte ratio PCE/PCE + NCE was counted in 400 bone marrow cells per slide, 800 cells per animal, totaling 4000 cells per treatment.

### 2.7. Statistical Analysis

To determine differences between treatments, data expressed as mean ± standard deviation were compared by one-way or two-way analysis of variance (ANOVA), followed by Tukey’s test (considering significant *p* values < 0.05) through the GraphPad 8 Prism program (Intuitive Software for Science, San Diego, CA, USA, version 7).

## 3. Results

### 3.1. Toxicological Evaluation

#### 3.1.1. Hippocratic, Behavioral, and Organ Weight Monitoring

During 11 weeks of DMBA treatment and 3 weeks (12th, 13th, and 14th) of drug therapy, in the different groups, no Hippocratic alterations were observed. Olive oil (10 mL/Kg), used as a vehicle for DMBA, did not induce behavioral changes in crossing, self-cleaning, or lifting, or in locomotor activity. Similar data were observed for CIT and LP-CIT at both concentrations. However, the animals treated with DMBA presented behavioral alterations in the rota rod, when compared to the vehicle, due to a reduction in permanence time (time in seconds) in the 1st week (169.00 ± 11.44 versus 180.00 ± 0.01); 5th week (1.694 ± 7.87 versus 180.00 ± 0.10); 6th week (168.87 ± 7.15 versus 180.00 ± 0.05); 7th week (168.57 ± 7.14 versus 180.00 ± 0.20); and 11th week (165.00 ± 8.03 versus 180.00 ± 0.03). Changes induced by DMBA were also observed as an increased number of falls for the 11th week (1.85 ± 1.51 versus 0.00). CPA, during the 14th week, increased lifting significantly when compared to the vehicle (32.00 ± 6.32 versus 14.00 ± 4.55) (Table 1).

During therapy with DMBA, CPA, CIT, and LP-CIT, no significant differences were observed for kidney weight. However, changes in liver weight were observed for DMBA when compared to the vehicle. In addition, there were significant changes regarding weight in the right and left breasts treated with DMBA, in relation to the vehicle group. Treatment with CPA, CIT, and LP-CIT, at both concentrations, reduced the weight of both breasts when compared to DMBA. CIT and LP-CIT showed more efficient results in tumor reduction, being more pronounced for LP-CIT at both concentrations. LP-CIT, in the two evaluated concentrations, presented no significant differences from CPA treatment (Figure 1).

#### 3.1.2. Biochemical and Hematological Profile

Free citrinin (CIT) and liposomal citrinin (LP-CIT) did not induce hematological changes in the red blood series, as observed after 11 weeks of DMBA tumor induction and 3 weeks of CIT therapy. No alterations were found for the white series, except for monocytes, when compared to data obtained for the vehicle and CIT and changes in platelets during CPA therapy compared to the vehicle (Table 2). However, CIT therapy induced liver alterations, due to a significant increase in glutamic-oxalacetic transaminase (GPT) and glutamic-pyruvic transaminase (GOT) compared to the control group. We did not observe alterations in the liver or kidneys of the animals under LP-CIT therapy (Table 3).

#### 3.1.3. Macroscopic and Histopathological Analysis in Mice with DMBA-Induced Breast Cancer

In the macroscopic analysis, the DMBA-induced tumor increased breast vascularity as well as inflammation and swelling of the breast tissue (Figure 2A). After LP-CIT therapy, at a dose of 6 µg/Kg, visibly reduced vascularization, inflammation, and breast swelling were observed (Figure 2B).

The histopathological profile from animals treated with DMBA-induced breast cancer showed characteristics of a carcinoma, with the presence of non-uniform ducts, ductal cell atypia, irregular nuclear contours, basophilia, and atypical ductal hyperplasia (Figure 3A–C). We also noted connective tissue surrounding the lesion—suggesting an invasive ductal carcinoma. The cells formed solid cords that infiltrated diffusely into the breast tissue. This is characteristic of an extensive desmoplastic reaction of the infiltrated tissue, with fibroblastic proliferation and collagen production, which gives the tumor a hard consistency on macroscopy. The tumor outline was poorly delimited, infiltrating the adipose tissue and healthy breast tissue. Moreover, cell division (mitosis), loss of the nucleus–cytoplasm relationship, and binucleation were observed in tumor cells.

In animals treated with CIT (2 μg/kg), no characteristics indicative of necrosis were observed. However, breast carcinoma characteristics remained, such as intraductal calcification, injured fibrous tissue with atypical ductal hyperplasia, and histopathological characteristics indicative of pre-neoplasia. Although the period of therapy with CIT (2 μg/kg) was only 3 weeks, which is a short period for an antineoplastic therapy, it was sufficient to identify significant changes in cell proliferation. In addition, many osteocytes, inflammatory cells, and atypical cell proliferation were identified (Figure 3D–F).

In animals treated with LP-CIT, breasts with a secretory system composed of lobules and ducts and located in the interlobular stroma formed by adipose tissue were observed. Lobules are circular structures surrounded by adipose tissue. Each lobe is made up of acini located in the interlobular stroma. The significant regression of typical carcinoma cell proliferation zones and features associated with tumor regression were observed at concentrations of 2 µg/kg and 6 µg/kg (Figure 3G–I).

### 3.2. Toxicogenetic Profile of DMBA, CPA, CIT, and LP-CIT

#### 3.2.1. Genotoxicity Induced by DMBA, CPA, CIT, and LP-CIT in Tumor Tissues and Healthy Cells of Female Mice

During BC induction and tumor treatment (14 weeks), DMBA (6 mg/kg per week) and cyclophosphamide (CPA, 25 mg/kg) induced genotoxic damage in breast cells and in non-neoplastic cells (liver and bone marrow), verified by significant increases in DI and FD when compared to the vehicle. Furthermore, CPA induced more genotoxicity in neoplastic cells than DMBA. CIT (2 μg/kg) and antineoplastic CPA induced genotoxicity after 3 weeks (12th, 13th, and 14th weeks) of therapy in breast carcinoma when compared to the vehicle. However, CIT showed lower genotoxic efficiency in these tumor cells when compared to CPA. When CPA and CIT were evaluated in bone marrow, except for the liver, CIT showed less genotoxic effects due to the significant differences in the damage index. Finally, at both concentrations of the nanoformulations (0.0 and 6.0 µg/kg), LP-CIT continued to cause DNA damage in breast neoplastic cells, with the highest concentration being statistically more efficient when compared to CIT. Treatment with 6.0 µg/kg LP-CIT was equally as genotoxic as CPA. In non-neoplastic cells (liver and bone marrow), LP-CIT (2.0 µg/kg) caused less genotoxic damage when compared to DMBA, CPA, and CIT, except for the liver, when compared to FD. Furthermore, LP-CIT (6.0 µg/kg) showed lower genotoxicity values than DMBA and CPA in all healthy cells in relation to DI; however, it was not less genotoxic than CIT (except for liver cells) (Table 4).

#### 3.2.2. Evaluation of DNA Damage Repair Capacity in Peripheral Blood Cells during and after Treatment with DMBA and Therapy with CPA, CIT, and LP-CIT

CPA, CIT, and LP-CIT (2 µg/mL and 6 µg/mL) induced genotoxicity due to significant increases in DI values (at three weeks for CPA and CIT, at week 13 for LP-CIT2, and at weeks 13 and 14 for LP-CIT6 and FD (in three weeks for CPA and only in the 13th week for all treatments) in blood cells when compared to baseline damage from DMBA tumor induction. However, in the study of induced DNA repair by DMBA, it was possible to verify that there was repair during CIT treatment and LP-CIT, observed by the statistically reduced values of the 14th and 15th weeks when compared to the 13th week, especially for DI. When CIT and the nanoformulations were compared to CPA, it was observed that CIT and the nanoformulations presented lower values of damage, being repaired during treatment. Conversely, CPA treatment did not present a repair capacity. Finally, the nanoformulations showed lower damage values compared to CIT in the same week of treatment (Figure 4).

#### 3.2.3. Cytotoxicity and Mutagenicity of CPA, CIT, and Nanoformulations in Breast, Liver, and Bone Marrow

In relation the application of the micronuclei test using breast, bone marrow, and liver cells, it was found that CIT, differently from CPA, did not induce clastogenic effects with micronuclei formation for bone marrow and liver cells. However, in the tumor cell analysis, it was observed that the highest concentration of the nanoformulation induced a significant increase in mutagenic effects, which was different from the results observed in free citrinin (CIT). Furthermore, the erythrocyte ratio showed that only CPA showed cytotoxic effects on the blood cells evaluated when compared to the vehicle (Table 5).

## 4. Discussion

Many efforts have been made to improve the selection process for natural chemical compounds with antitumor activity. The use of experiments in animal models and drug screening in studies in vitro are among the methods that have achieved notable success so far [27]. In this study, we evaluated the marine fungal metabolite known as citrinin, which was incorporated into liposomal nanosystems (LP-CIT). The LP-CIT parameters indicated that the formulation had a particle size, homogeneity, citrinin content, and characteristics that contributed to maintaining the stability of these carriers and it can be used for in vitro and in vivo tests.

DMBA is a polyaromatic hydrocarbon used in animal experiments [28]. During the carcinogenic process, DMBA metabolizes and generates various reactive metabolic intermediates that further form stable DNA adducts that are genotoxic and mutagenic [29]. Oxidative imbalance regulates protein and gene expression, which alters different signaling pathways and processes, such as angiogenesis, apoptosis, cell growth, DNA repair, and invasion [30,31]. Pantavos and collaborators [32] suggest that DMBA-induced breast cancer generates numerous reactive intermediates, such as OH, O_2_ and H_2_O_2_. After DMBA metabolization, reactive species potentiate oxidative stress by binding to the nucleophilic sites of cellular macromolecules, resulting in adenocarcinoma in the animal mammary gland comparable to human breast carcinoma [33].

DMBA induces changes in the levels of liver and kidney markers [34], although, in the present study, no hematological and biochemical changes were observed for DMBA and CPA. In the present study, DMBA (6 mg/kg per week), for 11 weeks, induced mammary carcinoma in virgin female mice, as observed by histopathological analyses. The mammary gland has, by nature, the property of accumulating fatty tissue, determining a specific location for DMBA to act [8,35].

We found that DMBA (6 mg/kg per week) induced behavioral toxicity (by increasing falls and lifting) in five weeks of an 11-week treatment for breast carcinoma induction, when compared to the vehicle (olive oil). CIT, LP-CIT, and cyclophosphamide (CPA) did not induce behavioral changes, indicating that CPA, CIT, and LP-CIT did not induce toxicity to the locomotor and behavioral system. However, as reported, no metabolic changes or changes in organ weight were observed, except for DMBA, which increased the liver and right and left breast weight. Drug treatment significantly reduced the organ weight, especially CIT and LP-CIT. These nanoformulations with citrinin significantly reduced the breast size when compared to free citrinin, showing greater antitumor efficiency when in nanosystems.

As observed, CIT and LP-CIT at the two evaluated concentrations reduced the mice tumor size; therapy with CIT proved to be effective, as it reduced tumor proliferation without inducing behavioral and locomotor effects and without significant weight changes in non-neoplastic organs. The best results for tumor regression were found with LP-CIT treatment, in both concentrations, as observed in the histological analysis.

Toxicity was observed for CIT (2 μg/kg) as an eosinophil and monocyte reduction. However, no changes were found in the red series. Likewise, there was no change in the hematological profile in the groups treated with LP-CIT. In terms of assessing hepatotoxicity, it is important to consider pharmacokinetic parameters, because the liver is an organ potentially capable of accumulating xenobiotics [36]. It is pertinent to point out that in previous studies with CIT, in high concentrations, it induced toxic effects to the liver, kidney, heart, and gastrointestinal tract [37,38,39,40].

In this study, CIT therapy, at a concentration of 2 µg/kg, induced liver alterations through significant increases in glutamic-oxalacetic transaminase (GPT) and glutamic-pyruvic transaminase (GOT), compared to the control group (vehicle). Such an effect of free citrinin can be justified by the production of bioactive metabolites that can affect hepatocytes during hepatic metabolism [41]. Conversely, LP -CIT maintained normal levels of liver and kidney markers at both concentrations tested.

As observed by Liu and collaborators [42], liposomes containing methotrexate (MTX) significantly inhibit tumor growth. Furthermore, liver histopathology and serum enzymes indicated that nanosystems markedly reduced hepatotoxicity compared to MTX alone. Many biologically active compounds, when used as drugs, have adverse effects on healthy organs or tissues, due to the unwanted systemic drug distribution, which is partially inactivated when reaching the target site [43]. However, the use of liposomes as drug carriers allows transport across membranes, providing selective passive tumor tissue targeting and reduced systemic toxicity [7].

Regarding genotoxic effects in tumor and non-tumor cells, DMBA induced genetic instability through DNA damage in breast tumor and peripheral blood cells. DMBA efficiently induces mammary tumors in animals by its effective genetic instability capacity [44,45]. In addition, DMBA also induces ROS formation and DNA damage in cells lacking the *BRCA1* gene, responsible for controlling the process of checking the mitotic cycle [46]. CPA, CIT, and LP-CIT induced genotoxicity in breast tumor cells and also in peripheral blood; however, LP-CIT (2 µg/mL) was less genotoxic in blood cells when compared to DMBA, CPA, and CIT. These results point out the efficiency of liposomal nanoformulations in minimizing the adverse effects of toxicity in non-tumor cells. CIT induces genotoxicity [47,48,49], chromosomal aberrations [47,50], and alterations in cell cycle [51], where oxidative stress is one of the main causes of damage [52,53].

The CIT polyketide has a free OH group on its carbons (C8 and C12), which, when associated with the mycotoxin ochratoxin, induces DNA damage via OH radical formation, generating oxidative damage [54]. CIT also stimulates the production of superoxide ions in the respiratory chain, promoting cell death by mitochondrial dysfunction [55] and by activating the MAPK signaling pathway [56].

Several studies present citrinin as an efficient cytotoxic agent in experimental animal models, including rabbits, rodents, and chickens [57,58,59,60]. Oliveira-Filho and collaborators [18] observed cell death, by nuclear fragmentation, with the comet test of mammary tumor tissue from female mice. CIT induced more apoptosis than DMBA, as well as CPA, at a concentration of 2 µg/kg [21]. One of the mechanisms reported to induce apoptosis is through blocking the cellular electron transport system, deregulating calcium homeostasis, and inhibiting intracellular signal transduction [54].

In the present study, DNA repair capacity was observed only against damage induced by CIT and LP-CIT in lymphocytes. DNA repair capacity and genetic instability are important events in oncological therapy, especially because chemotherapy agents and radiotherapy can induce chromosomal alterations [61]. The data from the micronucleus test in breast, bone marrow, and liver cells showed that CIT did not induce clastogenic effects with micronuclei formation in bone marrow and liver cells. The opposite was observed for CPA, as the carcinogenic capacity of many chemotherapeutic agents is partially dependent on their ability to induce mutagenic and clastogenic damage to DNA, including DNA adducts, replication errors, breaks, and crosslinks [62].

A good candidate for antitumor drugs should have the ability to induce cytotoxic, genotoxic, and mutagenic effects in neoplastic cells, blocking tumor progression. CIT can cause clastogenic effects in both in vivo and in vitro test systems [14]. Moreover, CIT induced apoptosis and cytotoxicity in tumor and non-tumor cells, especially in breast carcinoma cells. The DNA damage observed, in addition to its extension, can also be associated with the release of enzymes that act directly in the process of apoptosis and necrosis, such as the pro-apoptotic caspases-8, -9, and -3 and Bcl2 [63].

In another study using yeast cells, CIT (400 µg/mL) induced the strong expression of 68 genes related to oxidative stress, suggesting toxicity triggered by CIT at high concentrations [64]. De Oliveira-Filho and collaborators [65] observed that citrinin can exert its toxic, cytotoxic, and mutagenic effects in yeast strains, possibly through oxidative stress induction. CIT’s cytotoxicity can be attributed to its role as an apoptosis inducer. Numerous physical factors and chemical treatments can trigger cellular apoptotic signaling cascades by increasing intracellular ROS formation [66].

The DMBA-induced breast carcinoma model proved to be effective in inducing breast carcinoma. In addition, CIT and LP-CIT therapy was able to reduce tumor proliferation without inducing behavioral and locomotor changes, as well as weight changes in healthy organs and tissues. The antitumor effects of potential chemotherapeutic drugs such as citrinin have the ability to cause DNA damage through free radical formation, the alkylation of the DNA molecule, the inhibition of enzymes related to cell division, such as topoisomerases, and the inhibition of DNA repair [67].

The study by Salah and collaborators [68] investigated the mechanisms of action of citrinin in vitro. The data demonstrated that CIT significantly decreased the number of viable human intestinal HCT116 cells and induced apoptosis, including mitochondrial outer membrane permeabilization, caspase-3 activation, elevated ROS formation, and the loss of plasma membrane integrity. Furthermore, the genetic deficiency of the pro-apoptotic protein Bax protected cells against CIT-induced apoptosis, indicating that Bax is required for CIT-mediated toxicity. It was also found that CIT triggered endoplasmic reticulum (ER) stress.

In cancer chemotherapy, targeting drugs to solid tumor tissue using nanoparticles is one of the most promising treatments for specific delivery and safety [69]. The efficiency of liposomes as a drug delivery system in the pharmaceutical industry lies in their composition, which makes them biocompatible and biodegradable [70]. In addition, liposomes have the advantage of increasing stability through encapsulation, increasing the efficacy and therapeutic index of drugs, improving pharmacokinetic effects (i.e., decreasing clearance and increasing the circulation life), and reducing toxicity and genotoxicity [71,72].

When considering nanosystems for cancer treatment, the enhanced permeability and retention effect (EPR) plays a central role. The level of unwanted systemic drug release and intra-tumoral allocation, as well as the drug amount and kinetics, will also determine the ultimate efficacy of the treatment [73]. Thus, we have presented an interesting alternative to improve the antitumor activity of natural chemical compounds such as citrinin, which have cytotoxic effects on tumor cells but also have certain toxicity in healthy organs or non-neoplastic cells, which can be minimized with incorporation into liposomal nanosystems.

## 5. Conclusions

Citrinin showed antitumor activity by decreasing the breast tumor weight, improving histopathological markers, and promoting genotoxicity in tumor cells. However, CIT presented significant hepatotoxic and genotoxicity in non-tumor cells, which was reduced by LP-CIT, which maintained the antitumor activity and more efficiently modulated the toxicity and genotoxicity.

## Figures and Tables

**Figure 1 pharmaceutics-16-00174-f001:**
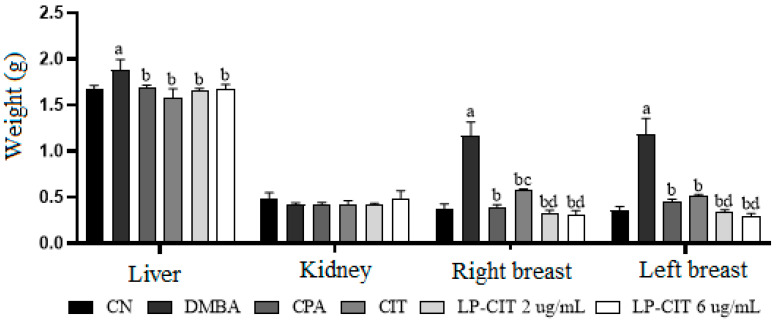
Organ weight profile of female mice after 11 weeks of treatment with DMBA and drug therapy (12th, 13th, and 14th weeks). Values represent mean ± standard deviation (*n* = 5); citrinin 2 μg/kg. DMBA = 7,12-dimethylbenzanthracene 6 mg/kg. CPA = cyclophosphamide 25 mg/kg. a, *p* < 0.05 compared to vehicle (olive oil), b, *p* < 0.05 compared to DMBA group, c, *p* < 0.05 compared to CPA group (ANOVA, two-way, Tukey post-test), d, *p* < 0, 05 compared to CIT group.

**Figure 2 pharmaceutics-16-00174-f002:**
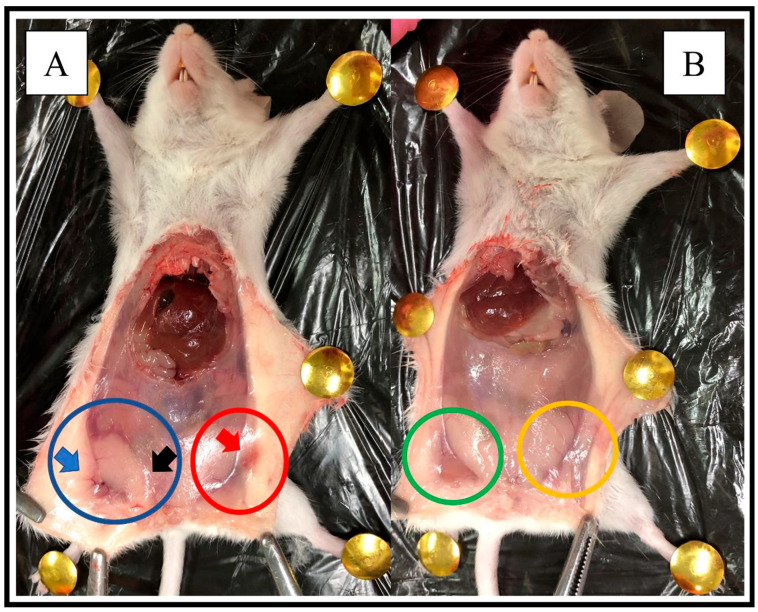
Macroscopic profile of the animals’ breasts, after DMBA-induced tumor. (**A**): DMBA treatment. Blue circle: right breast. Red circle: left breast. Blue arrow: indication of greater vascularization. Black and red arrow: breast swelling. (**B**): LP-CIT treatment: observation of breast size reduction after LP-CIT treatment (6 µg/Kg). Green circle: right breast. Yellow circle: left breast.

**Figure 3 pharmaceutics-16-00174-f003:**
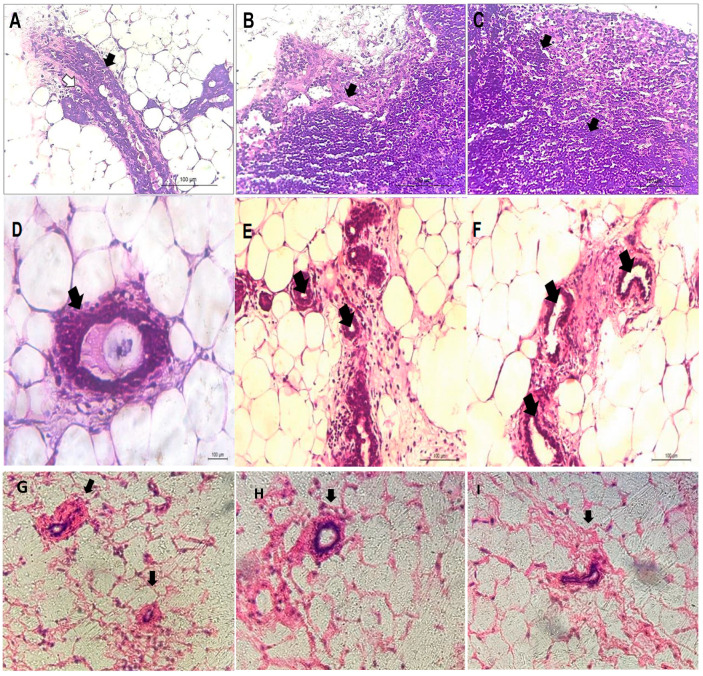
Histopathological profile of the right breast of female mice, after 11 weeks of DMBA treatment (**A**–**C**) and 3 weeks of therapy with free citrinin (**D**–**F**) and citrinin liposomes (LP-CIT) (**G**–**I**). (**A**): Atypical ductal hyperplasia (black arrow); cell with cytoplasm rich in glycogen (white arrow). (**B**): Connective tissue mixed with lesion, suggesting invasive carcinoma. (**C**): basophilia. (**D**): Usual ductal hyperplasia. (**E**): Intraductal calcification. (**F**): Atypical ductal hyperplasia. (**G**): Duct located in the intralobular stroma formed by adipose tissue. (**H**): Lobes with regular outline. (**I**): Significant tumor regression. (**H**,**E**): Stain (200× magnification).

**Figure 4 pharmaceutics-16-00174-f004:**
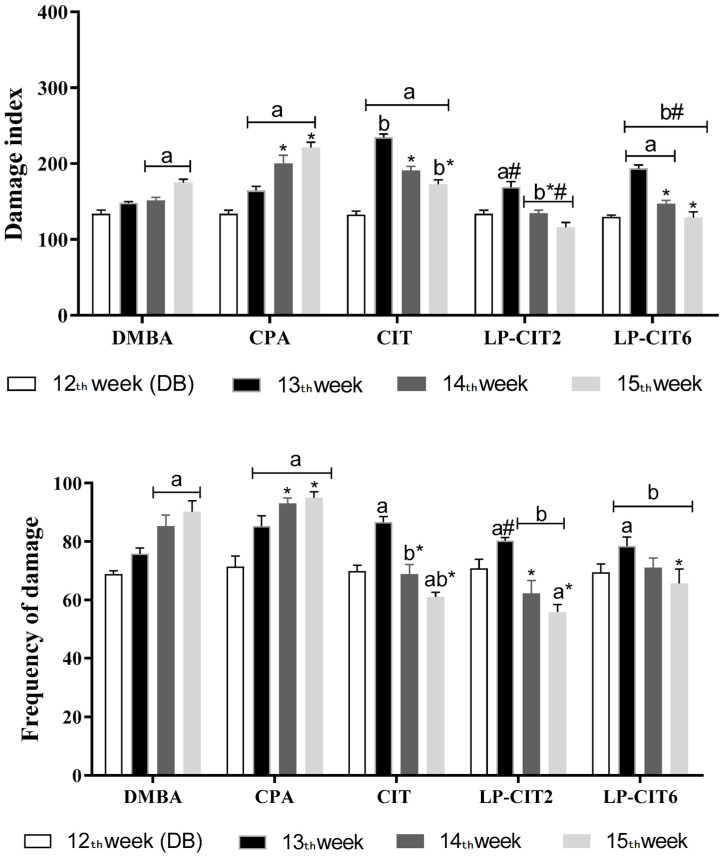
Study of DNA damage repair, in nucleated peripheral blood cells of female mice, during DMBA-induced breast cancer and therapy with CPA, isolated CIT, and LP-CIT (2 and 6 ug/mL). Damage index (0–400) and frequency of damage (0–100). Values represent mean ± standard deviation (*n* = 5). ANOVA, two way, Tukey’s post-test. a, *p* < 0.05 compared to baseline damage of DMBA in the same treatment, b, *p* < 0.05 compared to CPA in the same week; * *p* < 0.05 compared to the 13th week in the same treatment; # *p* < 0.05 compared to free CIT in the same week.

**Table 1 pharmaceutics-16-00174-t001:** Behavioral evaluation and locomotor activity of virgin female mice during 14 weeks of treatment with olive oil, 7,12-dimethylbenzanthracene (DMBA), and cyclophosphamide (CPA), free citrinin (CIT), and liposomal citrinin (LP-CIT).

Treatment Groups and Weeks		Behavioral Parameters and Locomotor Activity				
		Open Field			Rota Rod	
		Crossings	Self-Cleaning	Elevation	TP (s)	NQ
Olive oil	S 01	83.20 ± 20.60	5.00 ± 2.14	12.00 ± 6.16	180.00 ± 0.01	0.40 ± 0.02
	S 01	83.20 ± 20.60	5.00 ± 2.14	12.00 ± 6.16	180.00 ± 0.01	0.40 ± 0.02
	S 02	81.00 ± 16.80	14.40 ± 3.05	24.00 ± 7.48	174.40 ± 7.43	1.40 ± 0.89
	S 03	83.60 ± 33.40	11.40 ± 5.12	24.00 ± 19.5	180.00 ± 0.01	0.00 ± 0.00
	S 04	57.20 ± 9.44	14.00 ± 2.91	16.80 ± 2.16	176.80 ± 7.15	0.00 ± 0.00
	S 05	55.00 ± 14.80	12.60 ± 6.10	12.80 ± 6.26	180.00 ± 0.10	0.00 ± 0.00
	S 06	52.20 ± 21.40	9.80 ± 4.08	13.60± 8.05	180.00 ± 0.05	0.00 ± 0.00
	S 07	37.00 ± 26.40	8.40 ± 11.3	14.60 ± 4.55	180.00 ± 0.20	0.00 ± 0.00
	S 08	73.10 ± 15.00	7.70 ± 3.45	23.90 ± 5.46	180.00 ± 0.01	0.00 ± 0.00
	S 09	80.00 ±16.80	12.35 ± 5.23	27.74 ± 7.5	180.00 ± 0.03	0.00 ± 0.00
	S 10	82.30 ± 31.40	13.00 ± 4.42	22.41 ± 5.54	180.00 ± 0.02	0.00 ± 0.00
	S 11	56.50 ± 12.00	8.99 ± 2.34	23.60 ± 7.55	180.00 ± 0.03	0.00 ± 0.00
	S 12	55.00 ± 14.00	9.13 ± 3.51	21.80 ± 4.78	180.00 ± 0.10	0.00 ± 0.00
	S 13	55.20 ± 20.10	13.24 ± 5.1	18.59 ± 5.55	180.00 ± 0.10	0.00 ± 0.00
	S 14	39.00 ± 27.00	11.25 ± 5.12	14.00 ± 4.55	180.00 ± 0.02	0.00 ± 0.00
DMBA 6 mg/kg	S 01	68.20 ± 23.0	9.60 ± 4.82	20.60 ± 5.94	169.00 ± 11.44 ^a^	1.40 ± 0.67
	S 02	82.00 ± 30.0	15.80 ± 6.14	27.20 ± 4.60	176.00 ± 6.92	0.40 ± 0.28
	S 03	65.00 ± 14.6	11.20 ± 1.30	19.60 ± 3.43	180.00 ± 0.00	0.00 ± 0.00
	S 04	76.80 ± 11.3 ^a^	15.80 ± 7.59	19.40 ± 10.6	176.60 ± 7.60	0.60 ± 0.12
	S 05	76.40 ± 18.9	15.40 ± 13.9	20.40 ± 4.03	169.40 ± 7.87 ^a^	0.80 ± 0.32
	S 06	62.20 ± 24.3	11.40 ± 4.72	26.60 ± 12.1	168.87 ± 7.15 ^a^	0.00 ± 0.00
	S 07	60.80 ± 7.19	14.70 ± 10.3	19.80 ± 7.08	168.57 ± 7.14 ^a^	0.00 ± 0.00
	S 08	63.00 ± 5.80	10.30 ± 4.50	23.70 ± 7.30	180.00 ± 0.10	0.00 ± 0.00
	S 09	70.50 ± 6.20	12.34 ± 7.00	18.50 ± 5.20	180.00 ± 0.20	0.00 ± 0.00
	S 10	69.40 ± 3.90	13.43 ± 3.70	19.35 ± 4.30	180.00 ± 0.10	0.00 ± 0.00
	S 11	80.20 ± 14.00	15.37 ± 5.35	20.30 ± 5.34	165.00 ± 8.03 ^a^	1.85 ± 1.51 ^a^
CPA 25 mg/kg	S 12	54.67 ± 18.23	18.26 ± 6.24	22.6 ± 5.55	165.0 ± 15.00	4.30 ± 1.12
	S 13	78.00 ± 22.39	17.78 ± 4.78	21.8 ± 4.13	164.9 ± 13.50	1.80 ± 0.21
	S 14	68.28 ± 18.34	20.34 ± 9.73	32.00 ± 6.32 ^a^	177.73 ± 12.24	1.80 ± 0.19
CIT 2 μg/kg	S 12	50.28 ± 15.51	14.0 ± 4.96	24.28 ± 6.77	175.57 ± 9.14	0.71 ± 1.46
	S 13	51.14 ± 13.65	15.14 ± 4.87	23.71 ± 6.52	179.14 ± 2.23	0.42 ± 1.11
	S 14	47.14 ± 31.60	13.42 ± 8.59	17.42 ± 10.81	178.14 ± 3.07	0.28 ± 0.47
LP-CIT 2 μg/kg	S 12	82.30 ± 21.70	13.00 ± 4.42	22.41 ± 5.54	180.00 ± 0.02	0.00 ± 0.00
	S 13	73.10 ± 11.00	7.70 ± 3.45	24.90 ± 5.46	180.00 ± 0.20	0.00 ± 0.00
	S 14	80.00 ±13.80	12.35 ± 5.23	27.74 ± 7.5	180.00 ± 0.10	0.00 ± 0.00
LP-CIT 6 μg/kg	S 12	78.50 ± 7.40	12.54 ± 7.00	18.30 ± 5.30	180.00 ± 0.10	0.00 ± 0.00
	S 13	79.40 ± 2.90	14.43 ± 5.70	20.35 ± 4.30	180.00 ± 0.03	0.00 ± 0.00
	S 14	79.20 ± 4.00	16.34 ± 3.37	22.50 ± 6.34	180.00 ± 0.20	0.00 ± 0.00

Values represent mean ± standard deviation (*n* = 5). CIT = citrinin. DMBA = 7,12-dimethylbenzanthracene. CPA = cyclophosphamide. LP-CIT = citrinin-containing liposomes. TP = residence time. NQ = number of falls. S = treatment week. ANOVA (two-way), Tukey’s post-test. a, *p* < 0.05 compared to olive oil, in the same week.

**Table 2 pharmaceutics-16-00174-t002:** Hematological profile in mice with DMBA-induced breast cancer.

			Groups			
Hematological Profile	Vehicle 10 mg/kg	DMBA 6 mg/kg	CPA 25 mg/kg	CIT 2 mg/kg	LP-CIT 2 mg/kg	LP-CIT 6 mg/kg
Red series						
HEM (10^6^/uL)	7.5 ± 0.63	9.65 ± 0.32	9.97 ± 0.54	9.0 ± 0.70	12.21 ± 0.91	8.75 ± 2.17
HGB (g/dL)	14.12 ± 0.67	14.8 ± 0.42	15.50 ± 1.16	15.30 ± 1.40	15.52 ± 0.88	14.1 ± 0.48
HCT (%)	37.7 ± 2.95	54.1 ± 0.03	56.50 ± 2.86	51.56 ± 3.10	52.87 ± 1.54	45.0 ± 2.03
White series						
Leukocytes (10^3^/uL)	5.000 ± 1894.72	2.600 ± 424.26	3398.0 ± 840.70	2834 ± 427.20	2.908 ± 378.18	5.137 ± 1876.51
Neutrophils (%)	9.0 ± 1.87	4 ± 4.25	8.76 ± 2.51	11.80 ± 0.70	3.5 ± 2.64	17.25 ± 5.73
Lymphocytes (%)	83.0 ± 2.23	87.5 ± 1.92	77.8 ± 4.62	86.00 ± 5.90	85.75 ± 5.11	79.0 ± 7.87
Monocytes (%)	8.0 ± 1.87	4.5 ± 6.36	12.8 ± 4.21	1.20 ± 2.20 *	4.75 ± 3.59	2.25 ± 2.5
Eosinophils (%)	0 ± 0	0 ± 0	0.14 ± 0.05	0.10 ± 0.04	0 ± 0	0.5 ± 0.57
Others						
Platelets (10^9^/uL)	6672.00 ± 693.88	7850.00 ± 108.07	1034.00 ± 100.7 *	8531.0 ± 109.1	6390.0 ± 378.15	7985.0 ± 937.00

Values represent mean ± standard deviation (*n* = 5). CIT = citrinin. DMBA = 7,12-dimethylbenzanthracene. CPA = cyclophosphamide. HEM = red blood cells. HGB = hemoglobin. HTC = hematocrit. MCV = mean corpuscular volume. HCM = mean corpuscular hemoglobin. MCHC = mean corpuscular hemoglobin concentration. ANOVA (two-way), Tukey’s post-test. * *p* < 0.05 compared to vehicle.

**Table 3 pharmaceutics-16-00174-t003:** Biochemical profile for renal and hepatic enzymes after citrinin (CIT) and liposomal citrinin (LP-CIT) therapy in female mice with DMBA-induced breast cancer.

Biochemical Profile	Vehicle 10 mg/kg	DMBA 6 mg/kg	CPA 25 mg/kg	CIT 2 mg/kg	LP-CIT 2 mg/kg	LP-CIT 6 mg/kg
Urea (mg/dL)	28.2 ± 4.44	40 ± 1.41	36.0 ± 8.71	38.7 ± 9.40	28.25 ± 6.99	24.25 ± 1.25
Creatinine (mg/dL)	0.42 ± 0.13	0.25 ± 0.07	0.46 ± 0.05	0.50 ± 0.05	0.5 ± 0.18	0.475 ± 0.24
TGO (IU/L)	102.2 ± 13.63	142 ± 0.03	101.6 ± 13.6	231.5 ± 22.00 *	138 ± 18.97	105.75 ± 27.84
TGP (IU/L)	85.8 ± 9.52	101 ± 0.04	156.6 ± 6.65 *	186.5 ± 24.60 *	125.75 ± 9.53	120 ± 20.54

Mean ± standard deviation (*n* = 4). LP = empty liposome. CIT = citrinin. LP-CIT = liposomal citrinin. TGO = glutamic oxaloacetic transaminase. TGP = glutamic-pyruvic transaminase. * *p* < 0.05 compared to the control group.

**Table 4 pharmaceutics-16-00174-t004:** Genotoxicity (damage index and frequency of damage) in neoplastic cells (breast) and healthy cells (liver and bone marrow) of female mice, after 14 weeks of tumor induction and treatment.

Treatment	Dose		Damage Index			Frequency of Damage	
		Breast	Liver	Bone Marrow	Breast	Liver	Bone Marrow
Vehicle	10 mL/kg	24.60 ± 3.1	25.1 ± 10.9	34 ± 13.4	27.7 ± 3.9	32.2 ± 9.1	38.1 ± 10.2
DMBA	6 mg/kg	131.2 ± 9.2 ^a^	191.1 ± 18.7 ^a^	268.6 ± 18.7 ^a^	89.1 ± 2.2 ^a^	86.7 ± 3.5 ^a^	94.7± 2.4 ^a^
CPA	25 mg/kg	187.6 ± 6.8 ^ab^	166.4 ± 18.9 ^a^	155.9 ± 17.6 ^ab^	98.8 ± 2.9 ^ab^	96.3 ± 3.6 ^a^	96.4 ± 3.2 ^a^
CIT	2.0 µg/kg	135.4 ± 10.1 ^ac^	130.3 ± 22.7 ^ab^	93.8 ± 25.6 ^abc^	85.5 ± 4.2 ^ac^	81.3 ± 7.1 ^a^	82.1 ± 5.8 ^abc^
LP-CIT	2.0 µg/kg	150.1 ± 14.1 ^ac^	64.2 ± 10.2 ^abcd^	42.9 ± 8.9 ^bcd^	95.4 ± 4.2 ^a^	69.1 ± 4.9 ^abc^	46.8 ± 5.4 ^bcd^
	6.0 µg/kg	183.8 ± 10.2 ^abd^	82.9 ± 9.9 ^abcd^	55.9 ± 10.2 ^bc^	96.3 ± 3.6 ^ad^	76.9 ± 5.8 ^ac^	69.8 ± 5.9 ^abc^

Values were expressed as mean ± standard deviation using two-way ANOVA with Tukey’s test, having values with *p* < 0.05: a different when compared to the vehicle (olive oil); b different when compared to DMBA; c different when compared to CPA; d different from LP-CIT when compared to CIT.

**Table 5 pharmaceutics-16-00174-t005:** Cytotoxicity and mutagenicity of 3 weeks of therapy with CPA, CIT, and LP-CIT in breast neoplastic and non-neoplastic cells (bone marrow and liver cells) after 11 weeks of DMBA-induced breast cancer.

Group	Dose	Breast	Bone Marrow	Liver	PCE/EPC + NCE
Vehicle	10 mL/kg	6.1 ± 2.6	6.0 ± 1.3	5.5 ± 1.5	0.78 ± 0.05
CPA	25 mg/kg	39.4 ± 5.4 ^a^	28.4 ± 3.9 ^a^	34.6 ± 4.6 ^a^	0.46 ± 0.02 ^a^
CIT	2.0 µg/kg	12.0 ± 4.5 ^b^	10.0 ± 1.5 ^b^	9.2 ± 3.1 ^b^	0.73 ± 0.04 ^b^
LP-CIT	2.0 µg/kg	13.1 ± 4.8 ^b^	8.4 ± 3.5 ^b^	7.9 ± 3.7 ^b^	0.72 ± 0.03 ^b^
	6.0 µg/kg	24.4 ± 3.5 ^abc^	10.2 ± 4.7 ^b^	9.2 ± 4.0 ^b^	0.71 ± 0.06 ^b^

Values represent mean ± standard deviation of MN numbers: micronuclei; a, *p* < 0.05 compared to olive oil group, b, *p* < 0.05 compared to CPA, c, *p* < 0.05 compared to CIT (two-way ANOVA, Tukey post-test). Relationship between the number of polychromatic erythrocytes and the sum of polychromatic (PCE) and normochromatic (NCE) erythrocytes in 400 analyzed cells (PCE/PCE + NCE).

## Data Availability

Data are contained within the article.

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
