# Peer review of "Advances in Antitumor Effects Using Liposomal Citrinin in Induced Breast Cancer Model"

_pharmaceutics, 2024, doi:10.3390/pharmaceutics16020174_

Round 1

Reviewer 1 Report

Comments and Suggestions for Authors

In this manuscript, considering the constant search for novel antitumor drugs and better delivery systems, the present study aimed to incorporate the fungal metabolite citrinin into liposomal nanosystem, evaluating its toxic, genotoxic and antitumor effects in an animal model of breast cancer induced by dimethylbenzanthracene. This manuscript is interesting and can be published after minor revsion.

1. LP-CIT presented pH 7.4, particle diameter of 143.3 nm, polydispersion index (PDI) of 0.3 and zeta potential of +28.5 mV. Please explain why it possessed positive surface charge.

2. During therapy with DMBA, CPA, CIT and LP-CIT no significant differences were observed for kidney weight. Please explain what is the aim for studing kidney weight.

3. In page 2, line 47-48, "Nanoparticles present ...their volume counterparts."  It is better to describe the nanomedicine rather than describing the nanoparticles because researchers have already been familiar with the feature of nanoparticles. For describing nanomedicine, the following nanomedicine literatures could be updated,DOI:10.3390/pharmaceutics14071522; DOI:10.1016/j.actbio.2023.01.022

4. Furthermore, the erythrocyte ratio showed that only CPA showed cytotoxic effects on blood cells evaluated when compared to the vehicle. Please expain why CPA showed cytotoxic effects on blood cells.

Author Response

"Please see the attachment".

Reviewer 2 Report

Comments and Suggestions for Authors

This is an interesting study and may be considered for publication after minor revision.

1-The histological investigation is not sufficient to confirm the therapeutic effects of prepared nano-formulation, they need to do Caspase 3, 9 or BCl-2 staining on the tissues to show the anticancer effect of the nano-formulation.

2- Why they selected to examine behavioral parameters such as open field and rota rod which is mainly used in neurological conditions, please explain with justification.     

Comments on the Quality of English Language

Minor editing
